# Societal cost of nine selected maternal morbidities in the United States

**Sasigant So O'Neil**[1]*, **Isabel Platt**[1], **Divya Vohra**[1], **Emma Pendl-Robinson**[1], **Eric Dehus**[1], **Laurie Zephyrin**[2], **Kara Zivin**[1]

1 Mathematica, Princeton, New Jersey, United States of America, 2 The Commonwealth Fund, New York, New York, United States of America

* soneil@mathematica-mpr.com

## Abstract

### Objective

To estimate the cost of maternal morbidity for all 2019 pregnancies and births in the United States.

### Methods

Using data from 2010 to 2020, we developed a cost analysis model that calculated the excess cases of outcomes attributed to nine maternal morbidity conditions with evidence of outcomes in the literature. We then modeled the associated medical and nonmedical costs of each outcome incurred by birthing people and their children in 2019, projected through five years postpartum.

### Results

We estimated that the total cost of nine maternal morbidity conditions for all pregnancies and births in 2019 was $32.3 billion from conception to five years postpartum, amounting to $8,624 in societal costs per birthing person.

### Conclusion

We found only nine maternal morbidity conditions with sufficient supporting evidence of linkages to outcomes and costs. The lack of comprehensive data for other conditions suggests that maternal morbidity exacts a higher toll on society than we found.

### Policy implications

Although this study likely provides lower bound cost estimates, it establishes the substantial adverse societal impact of maternal morbidity and suggests further opportunities to invest in maternal health.

**Data Availability Statement:** All relevant data are within the paper and its Supporting information files.

**Funding:** The Commonwealth Fund (https://www.commonwealthfund.org/) supported this work under contract number 20212979, received by SO. The funder had no role in study design, data collection, or in the analysis of results, and this article does not necessarily reflect the funder's views or opinions. However, the funder provided a critical reading of the manuscript and offered suggestions for revisions.

**Competing interests:** The authors have declared that no competing interests exist.

## Introduction

Maternal morbidity encompasses multiple physical and psychological health conditions that result from or are aggravated by pregnancy [1]. These conditions can start during pregnancy or within a year after delivery. The long-term effects, however, can last months or years, ranging from short acute episodes to longer chronic ailments [2]. These conditions do not necessarily lead to maternal mortality, but they can negatively affect quality of life. The Centers for Disease Control and Prevention identifies 21 severe maternal morbidity indicators, such as blood transfusion, eclampsia, hysterectomy, and sepsis, using hospital discharge data from delivery [1]. The occurrence of severe maternal morbidity has approximately doubled over a 15-year period, affecting 1.4% of birthing people during delivery in the United States [1, 3, 4].

Maternal mortality is the most serious consequence of maternal morbidity. The United States has a maternal mortality ratio of 20 deaths per 100,000 pregnancies, the worst of high-income countries [5]. Other disabilities and chronic illness stemming from maternal morbidity—often referred to as near misses—can have ongoing and compounded effects on a birthing person, their children, and other household members, shaping their workforce participation, nutrition, schooling, and other factors affecting quality of life [6–8]. Maternal morbidity can lead to adverse outcomes for birthing people, such as cesarean delivery and stroke, and adverse outcomes for their children, such as asthma, preterm birth, and suboptimal breastfeeding [7, 9–14].

Because of the many conditions associated with maternal morbidity, few studies have attempted to comprehensively estimate its overall costs [15]. This paper estimates the cost of nine maternal morbidity conditions with strong underlying evidence linking them to pregnancies or live births in the United States. We follow each maternal–child pair in the 2019 birth cohort from pregnancy to five years postpartum to highlight near-term economic impacts likely to be most salient to policymakers. To our knowledge, this cost model represents the most comprehensive analysis to date of the economic costs of maternal morbidity in the United States.

This model estimates the costs for select conditions associated with maternal morbidity, though one should interpret results with caution. Our estimate could represent only a lower bound of overall costs because only nine conditions had sufficient information to link to outcomes and costs. At the same time, our results might overestimate certain costs because some birthing people may experience comorbidities (that is, complexities that are not captured in our model). Our model calculates an estimate of the combined independent costs of nine selected maternal morbidities, supplying initial evidence to inform policy.

Our study refers to anyone who has experienced a pregnancy as a "birthing person." We use the term "maternal morbidity" to describe the adverse medical conditions experienced by birthing people.

## Methods

This study used a similar approach to that of Luca et al. [16], who quantified the economic impact of untreated maternal mental health conditions (MMHCs) in the United States. We started with the CDC's list of 21 severe maternal morbidity indicators, which defines a subset of maternal morbidity conditions through diagnosis codes, and added ten other non-life threatening adverse perinatal medical and mental health conditions directly resulting from pregnancy based on expert recommendations [1]. Through the data extraction process, described in the literature review section below, we narrowed the scope of our analysis to nine maternal morbidity conditions with supporting evidence of subsequent outcomes and associated costs.

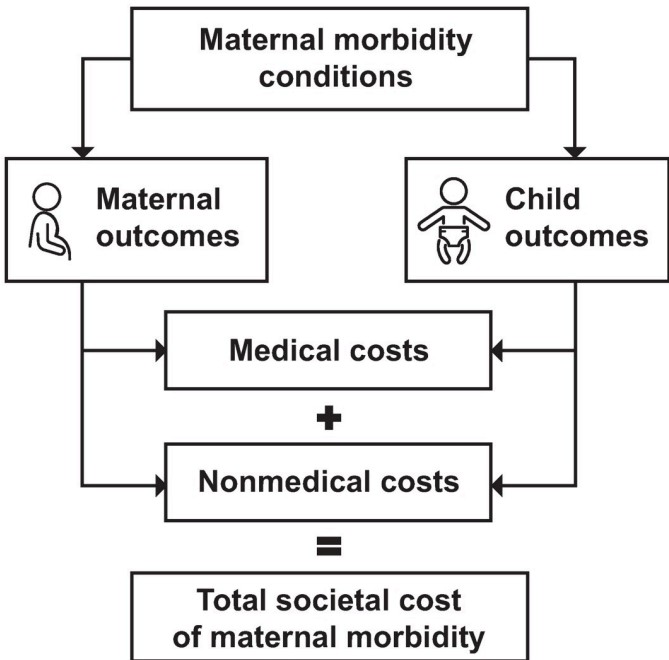

**Fig 1. Conceptual framework of our model that calculates the costs of outcomes attributed to nine maternal morbidity conditions, 2019 birth cohort.** Our model estimated the excess medical and nonmedical costs of maternal and child outcomes associated with maternal morbidity conditions.

We define societal cost as the combined estimated excess medical and nonmedical costs from maternal morbidity for birthing people and their children based on a conceptual model (Fig 1). Medical costs include those directly incurred through the health care delivery system, such as hospitalization costs for birthing people and their children. Nonmedical costs include costs incurred outside the health care delivery system such as productivity loss and absenteeism (that is, missing days of work). To avoid overestimating the risk of outcomes associated with maternal morbidity, our conceptual model focused on primary outcomes that are directly associated with maternal morbidity conditions. For example, infants who experience suboptimal breastfeeding—one of our modeled outcomes—have a higher risk of death from sudden infant death syndrome. We did not include the cost of this secondary outcome in our model because we already include sudden infant death syndrome as a primary outcome of MMHCs, one of the nine maternal morbidity conditions we studied.

### Literature review

To identify the maternal morbidity conditions and subsequent outcomes we used in our model, we conducted a comprehensive literature review following Preferred Reporting Items for Systematic Reviews and Meta-Analyses guidelines (Fig 2) [17]. We conducted three searches: (1) the prevalence or incidence of each maternal morbidity condition, (2) the likelihood of adverse medical and nonmedical outcomes associated with maternal morbidity, and (3) the associated medical and nonmedical costs of each outcome. We searched key databases to identify original articles and meta-analyses published in peer-reviewed journals, restricting the searches to articles published from 2010 to 2021. We supplemented these articles with grey literature and other reports focusing on the impacts and costs of maternal morbidity.

From the results of our database searches, we reviewed titles, abstracts, and full text to determine high-quality estimates for our model using the following criteria: (1) study used

**Identification of studies via databases, grey literature, and snowballing**

Duplicate records removed (n = 3,064)

Records identified from:
   Databases (n = 8,337;
   Search 1 = 177;
   Search 2 = 6,421;
   Search 3 = 1,739)
   Grey literature and
   snowballing (n = 95)

Abstracts excluded from review:
   Did not evaluate any of the desired associations
   (n = 2,975)
   Wrong population (n = 786)
   Findings focused on intervention or screening effects
   (n = 490)
   Study set in low- or middle-income country (n = 215)
   Study set in country with single-payer system (n = 164)
   Wrong study design (n = 124)
   Evaluated a risk factor for maternal morbidity (n = 119)
   Data more than 10 years old (n = 34)
   Unable to retrieve full text (n = 5)
   Background article (n = 4)

Abstracts screened
(n = 5,273)

Records excluded from article pool:
   Data more than 10 years old (n = 96)
   Did not evaluate any of the desired
   associations (n = 45)
   Did not provide estimates or results
   were not significant (n = 21)
   Background article (n = 12)
   Study set in country with single-payer
   healthcare system (n = 11)
   Evaluated a risk factor for maternal
   morbidity (n = 10)
   Findings focused on intervention or
   screening effects (n = 9)
   Wrong population (n = 6)
   Focused on biomarkers (n = 5)
   Insufficient controls (n = 5)
   Study set in low- or middle-income
   country (n = 4)
   Article not in English (n = 1)

Full-text records screened
(n = 449)

Records excluded from final model:
   Did not evaluate any of the desired associations (n = 41)
   Morbidity or outcome not part of conceptual model
   (n = 29)
   Mental health morbidity (n = 22)
   Study does not contain usable estimates (n = 20)
   Removed connection due to insufficient evidence (n = 18)
   Data more than 10 years old (n = 15)
   Wrong population (n = 14)
   Medical costs are from outside the U.S. (n = 9)
   Only contains pooled maternal morbidity estimates
   (n = 7)
   Data not from an OECD country (n = 4)
   Morbidity developed pre-pregnancy (n = 4)
   Outcome outside time frame (n = 4)
   Estimate not statistically significant (n = 3)
   Already had a U.S. estimate to support connection (n = 2)

Records assessed for
model inclusion
(n = 224)

Studies included in model
(n = 32)

Studies included in model
(n = 33)

**Fig 2. Title: Preferred reporting items for systematic reviews and meta-analyses flowchart of article selection for final model, using data primarily from 2010 to 2020.** This figure details the number of articles we found in each of our three searches, the number we selected for full-text review after screening title and abstracts, the number of articles we excluded from the model for each reason, and the final number of articles identified for model inclusion.

adequate controls to address confounding factors, (2) outcomes had direct associations with maternal morbidity conditions, (3) outcomes were quantifiable in monetary terms, and (4) incidence and prevalence estimates and medical costs came from the United States. To ensure that we modeled maternal morbidity–outcome connections with strong evidence, we included only connections with evidence in three or more studies and that had at least one cost estimate for the outcomes (either from the Healthcare Cost and Utilization Project's Nationwide Inpatient Sample or the literature) [18]. We narrowed our original 31 maternal morbidity conditions to nine for inclusion in the model, five of which represent severe maternal morbidity conditions. Details of the literature search strategy appear in S1 Appendix.

## Baseline rates

We obtained the baseline incidence or prevalence rates for maternal morbidity conditions using the most recent and relevant statistics available, relying primarily on the Healthcare Cost and Utilization Project's Nationwide Inpatient Sample. When estimates were unavailable through published reports using these data, we drew from other government agencies or peer-reviewed studies from our literature review. We used a similar process to identify the baseline

incidence or prevalence rates for the outcomes associated with maternal morbidity. When possible, we limited the population to pregnant and postpartum people and children through age five. S2 Appendix contains the maternal morbidity and outcome incidence and prevalence estimates we used in the main model.

## Cost estimates

To calculate the cost of maternal morbidity in the United States, we estimated the average incremental cost per birthing person or child for each outcome attributable to a maternal morbidity condition. Each outcome resulted in medical costs (such as a longer stay for a delivery hospitalization), nonmedical costs (such as lost wages), or some combination of them. We used the most recent and relevant estimates from the Agency for Healthcare Research and Quality, CDC, and other government agencies along with estimates from peer-reviewed articles from our literature review. Although the severity of each outcome can differ by person, we used the average estimates from the literature to populate our model. We standardized the costs to annual units and converted them to 2019 dollars using the medical component of the Consumer Price Index. S3 Appendix describes the sources we used to obtain cost estimates for maternal morbidity outcomes.

## Modeling

We created a model that estimated the costs of maternal morbidity among birthing people and their children for the 2019 birth cohort through five years postpartum, the period most salient to policymakers and fiscal funding time periods. Our key parameters included the following:

1. Number of pregnancies or live births in the United States in 2019, depending on whether the outcome from the condition occurred during pregnancy, at delivery, or postpartum

2. Prevalence of each maternal morbidity condition

3. Incidence of each outcome associated with a maternal morbidity condition

4. Impact estimates of each morbidity–outcome connection, such as the odds of a preterm birth given a maternal hemorrhage

5. Documented medical and nonmedical costs for each outcome (understanding that costs vary by severity of each condition or outcome)

6. Medical inflation rate and discount rate for all conditions, and remission rate for MMHCs

For each maternal morbidity condition associated with a given outcome, we identified the highest and lowest impact estimate from our literature search results as our range and used the midpoint from across the papers as the main estimate. If we identified only one estimate that met our inclusion criteria after further review of the studies, we used the 95% confidence interval as the high and low estimates. This strategy allowed us to include the full range of estimates meeting our criteria in the model. S4 Appendix contains the full set of impact estimates used in our modeling, and S5 Appendix contains the full set of model parameters.

Our model used the input parameters to calculate the medical and nonmedical costs of each morbidity–outcome connection, which reflect the number of birthing people or children who would experience an outcome directly because of a maternal morbidity condition. For each outcome, we used context from the literature to assume whether the costs occurred once (such as stillbirth) or on an ongoing basis (such as juvenile onset type 1 diabetes). S6 Appendix contains more detail on how we calculated the annual incremental excess costs of each outcome.

Following the U.S. Public Health Service's recommendations for cost models, we discounted costs at an annual rate of 3% to reflect the lower economic value of future expenses and accounted for increases in prices of healthcare services and commodities across time [19]. For medical costs, we used the medical care component of the Consumer Price Index to adjust prices to 2019 dollars. For non-medical costs, we used the Consumer Price Index for all items, less food and energy, to adjust prices to 2019 dollars [20]. We added the medical and nonmedical costs associated with each maternal morbidity condition to estimate the total societal costs of nine maternal morbidity conditions for the 2019 birth cohort through five years postpartum.

We conducted a series of deterministic sensitivity analyses to understand the sensitivity of our cost model to variation in prevalence, cost, and impact estimates, and we identified which model parameters had the greatest effect on changes to the total societal cost of maternal morbidity. We varied each incidence or prevalence rate individually using the high and low estimates to determine how each condition or outcome affected the total cost of maternal morbidity. Then, we varied each impact estimate individually. S7 Appendix contains more information on the sensitivity analyses we conducted.

## Results

In our literature review, we initially identified 8,337 studies through our database search and 95 papers through our review of the grey literature. After deduplication, the total number of unique records was 5,365. Using a systematic narrowing process (see S1 Appendix), we identified 32 studies that supported connections between nine maternal morbidity conditions and 24 maternal and child outcomes (Fig 3). To complete our data sources, we included 32 summaries or data briefs of publicly available data for the most recent prevalence and cost estimates and 44 studies from Luca et al. related to MMHCs [16]. Using the prevalence estimates for each of the nine conditions included in our model, we aggregated costs associated with each condition for all pregnancies or live births in 2019 to estimate a total cost of $32.3 billion from conception to five years postpartum. Of the total cost, we estimated $18.7 billion in medical costs and $13.6 billion in nonmedical costs. Two-thirds of these costs occurred within the first year postpartum.

MMHCs ($18.1 billion), hypertensive disorders ($7.5 billion), gestational diabetes mellitus ($4.8 billion), and hemorrhage ($1.8 billion) generated the largest costs, in part because these are the most prevalent conditions among those with documented cost information. Child outcomes accounted for about 74% of the total costs ($24.0 billion), and maternal outcomes accounted for about 26% ($8.3 billion). The specific child outcomes driving the costs of these conditions included preterm birth ($13.7 billion), developmental disabilities ($6.5 billion), and respiratory distress syndrome ($2.1 billion). The maternal outcomes with the highest costs included productivity loss ($6.6 billion), cesarean section delivery ($895 million), and increased peripartum stay ($350 million). These nine maternal morbidity conditions and outcomes amounted to an average of $8,624 in additional costs to society for each maternal–child pair of the more than 3.7 million births in the United States annually. Table 1 summarizes our findings, and our full results appear in S8 Appendix.

More than half the costs (58%) were borne by the medical system, and the rest (42%) were borne by employers and other nonmedical sectors. Nonmedical costs included losses in productivity ($6.6 billion), additional social service provision for behavioral and developmental disorders in children ($6.5 billion), and increased family enrollment in and use of social programs, such as the Supplemental Nutrition Assistance Program; the Special Supplemental Nutrition Program for Women, Infants, and Children; Medicaid; and Temporary Assistance for Needy Families ($239 million).

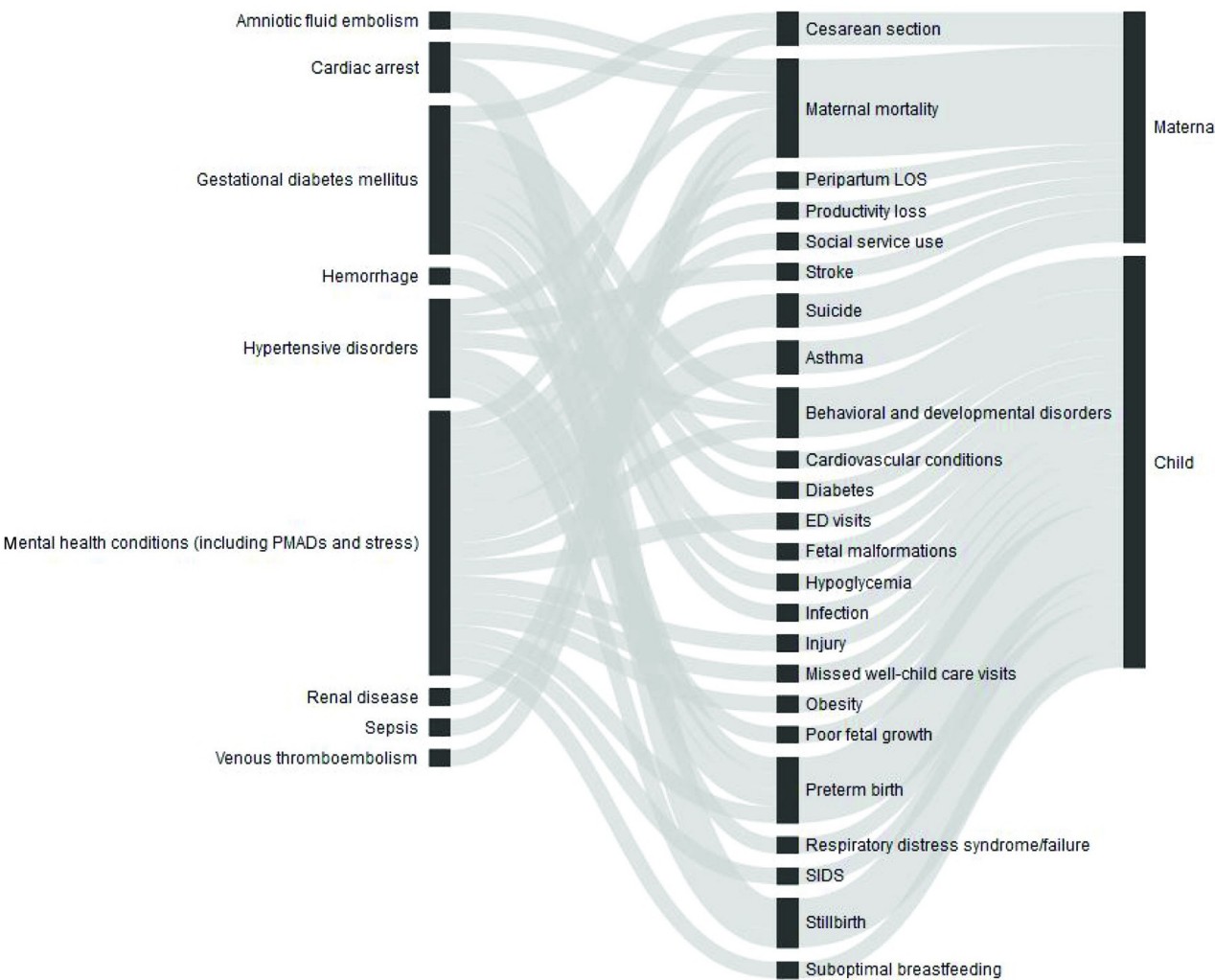

**Fig 3. Connections between nine maternal morbidity conditions and 24 outcomes, 2019 birth cohort.** This figure illustrates the connections included in our final model between nine maternal morbidity conditions and 24 outcomes. Each connection has a statistically significant evidence base or expert recommendation for inclusions and documented associated costs.

The sensitivity analyses showed that our model estimated a range of $12.1 billion to $57.6 billion by varying all prevalence and impact estimates at once. Preterm birth associated with MMHCs proved the most sensitive to variation––the incidence of preterm birth could decrease the total costs of maternal morbidity by up to $6 billion or increase them by up to $5 billion relative to our main model results. S5 Appendix provides the range of parameters we tested, and S7 Appendix shows the results of the sensitivity analyses in a tornado diagram.

## Discussion

Our findings demonstrate that maternal morbidity places a substantial economic toll on society. In particular, MMHCs, gestational diabetes, and hypertension have the highest costs, consistent with the results of Moran et al.'s systematic review of the incremental costs of maternal morbidity conditions [15]. In addition, our estimate far exceeds recent findings of Phibbs et al. and Chen et al., who projected an excess of $250 million and $630 million, respectively, in direct medical costs because of severe maternal morbidity [21, 22]. Our model projects

**Table 1. Cost estimates of maternal morbidity conditions in the United States for the 2019 birth cohort (in millions $).**

| Maternal morbidity condition | Total | Year 0 | Year 1 | Year 2 | Year 3 | Year 4 | Year 5 |
|---|---|---|---|---|---|---|---|
| Amniotic fluid embolism | 4.4 | 4.4 | 0.0 | 0.0 | 0.0 | 0.0 | 0.0 |
| Cardiac arrest | 10.9 | 10.9 | 0.0 | 0.0 | 0.0 | 0.0 | 0.0 |
| Gestational diabetes mellitus | 4,843.9 | 4,049.2 | 158.9 | 166.1 | 173.6 | 181.5 | 189.7 |
| Hemorrhage | 1,828.9 | 1,828.9 | 0.0 | 0.0 | 0.0 | 0.0 | 0.0 |
| Hypertensive disorders | 7,540.8 | 6,231.4 | 261.8 | 273.7 | 286.1 | 299.1 | 312.6 |
| Maternal mental health conditions | 18,059.0 | 9,782.6 | 1,655.0 | 1,730.0 | 1,808.4 | 1,890.3 | 1,975.9 |
| Renal disease | 3.0 | 3.0 | 0.0 | 0.0 | 0.0 | 0.0 | 0.0 |
| Sepsis | 3.3 | 3.3 | 0.0 | 0.0 | 0.0 | 0.0 | 0.0 |
| Venous thromboembolism | 6.4 | 6.4 | 0.0 | 0.0 | 0.0 | 0.0 | 0.0 |
| **Total** | **32,300.6** | **21,920.1** | **2,075.7** | **2,169.8** | **2,268.1** | **2,370.9** | **2,478.2** |

This table shows the total estimated costs of nine maternal morbidity conditions from conception to five years postpartum, estimated for the U.S. 2019 birth cohort. Maternal morbidity conditions with acute outcomes have one-time costs, while conditions with chronic outcomes have ongoing costs, modeled through five years postpartum. S8 Appendix contains the full results of the model by condition and outcome.

substantially higher costs by including the costs of outcomes associated with morbidity conditions—rather than limiting costs to the medical conditions themselves—and expanding the scope of the analysis to maternal morbidity conditions more broadly. The costs could rise if we extended our model beyond five-years postpartum (the period we considered most salient to policymakers), although most costs accrued during the first year.

We found evidence to support inclusion of nine maternal morbidity conditions in the cost model. Among all 2019 births, we estimated these nine conditions cost society $32.3 billion—with a lower bound of $12.1 billion and upper bound of $57.6 billion—from the beginning of pregnancy to five years postpartum. These significant costs show a need for further investments in evidence-based maternal health initiatives, such as midwifery models of care that extend beyond birth, comprehensive gender-specific primary health care that provides seamless transitions in and out of pregnancy, and community-based models of maternity care [23–26]. Initiatives should consider how social and structural factors beyond clinical care, such as unstable housing, lack of transportation, and racism, drive maternal outcomes [24, 27]. Policies that extend postpartum coverage or expand insurance coverage more broadly could incorporate these initiatives, and programming that incorporates a holistic approach to maternal health. It is important, however, to consider the implications of our findings in the context of the study limitations discussed here.

## Limitations

Our cost estimates for the nine maternal morbidity conditions are limited by the completeness of costs documented in the literature and the accuracy of model parameters. In addition, unaccounted-for interactions between the modeled conditions might amplify or moderate the costs of individual conditions. We also lack data for maternal morbidity conditions beyond the nine we selected and for costs by subgroups.

**Completeness of costs for nine selected conditions.** The preponderance of medical costs during the delivery period implies our estimates might have missed costs associated with longer-term consequences of maternal morbidity, such as associated chronic conditions that develop later for birthing people and their children, as well as stroke, educational challenges, or future earnings for children. In addition, our model does not account for costs of readmissions because of maternal morbidity, which could substantially increase our estimate [28, 29].

Furthermore, the literature contained few nonmedical costs for conditions other than MMHCs. Studies of other health conditions found that nonmedical costs from lost earnings, productivity loss, and other indirect costs can account for more than half the overall costs [30–32]. For our study, nonmedical costs accounted for 42% of total costs. These documented nonmedical costs mainly stem from MMHCs, the maternal morbidity condition for which we had the most complete outcomes and cost information. More information on nonmedical costs for the other maternal morbidity conditions could greatly increase our estimate of nonmedical and total costs. Finally, although we recognize that maternal morbidity can affect nonmaternal family members, such as sibling behavioral development, we only modeled the costs of the nine maternal morbidity conditions as they related to the birthing person and child.

**Accuracy of model parameters.** Our model uses data from secondary data and peer-reviewed literature that calculate statistical estimates to varying levels of statistical precision. In addition, the computer software used in this model introduced a slight rounding error. The precision could affect the accuracy of the societal cost and other statistical estimates reported.

**Unaccounted-for interactions and secondary outcomes.** To develop costs for each outcome, we modeled the likelihood of an outcome for each person with a given condition and the associated costs. Our model, however, cannot account for the effects of comorbidities. For example, a pregnant person with hypertension and gestational diabetes faces an increased risk for a preterm birth, and our model calculates only the independent costs of the individual conditions. These comorbidities could interact to increase or decrease the likelihood of a preterm birth beyond our impact estimate, resulting in an ambiguous effect on costs. In this example, the cost of delivery for a pregnant person with hypertension and gestational diabetes could end up substantially higher than the cost of delivery for a pregnant person with only one of these conditions. Despite the limitation that the model cannot account for comorbidities, the general lack of comprehensive data for other morbidities and outcomes suggests that maternal morbidity could have a much higher societal cost impact than we found.

Furthermore, to avoid overestimating the risk of outcomes associated with any given maternal morbidity condition, we chose not to model secondary outcomes. This approach might have led to our underestimating the costs of maternal morbidity. For example, we included child developmental disorders as one of the model outcomes associated with MMHCs. Many of these children might subsequently require use of social services, such as Supplemental Security Income, but we did not include this set of costs in the model because we already included the risk of social service use directly from MMHCs. Other researchers could revise the model to account for interaction effects of multiple conditions or secondary outcomes, which can provide a more comprehensive cost estimate for maternal morbidity overall.

**Lack of information for all maternal morbidity conditions and long-term costs, and by subgroup.** Our estimates likely do not capture the full costs of maternal morbidity for several reasons. First, although maternal morbidity includes many conditions and connections to outcomes, we could only find literature meeting our criteria to support modeling 32 connections between nine conditions and 24 outcomes (S1 Appendix). Second, we designed this model to focus on a six-year period (pregnancy to five years postpartum) so that stakeholders could understand the immediate impacts of maternal morbidity, especially as they consider allocating resources to interventions. We recognize that maternal morbidity can have long-term effects on the birthing person and the child, indicating that our estimates might represent only a fraction of the lifetime costs.

Ideally, we would have examined costs by various subgroups, such as race and ethnicity and Medicaid status, to understand the variation of cost burden for specific populations. For example, research has shown that Black birthing people have higher rates of severe maternal morbidity, which corresponds to a higher incidence of outcomes such as preterm birth and their

associated costs [33]. Further, maternal mental health conditions get screened and diagnosed at a higher rate among non-Hispanic whites than Hispanic or Black birthing people, contributing to even greater disparities in measuring the true cost burden among minoritized populations.

The literature did not provide sufficient evidence of morbidity-outcome connections by racial or ethnic subgroup for us to include in our model. Additional data and analyses documenting incidence and prevalence of maternal morbidity conditions and costs by subgroup would deliver a more complete understanding of populations that bear a disproportionate burden and experience inequities in structural and social factors that lead to disparities in outcomes. Such information can better support decision making on medical and non-medical interventions that could have the greatest impact on maternal health–related outcomes and costs.

## Conclusions

Today's case of maternal morbidity can result in tomorrow's maternal death, which means that better measurement and reporting of maternal morbidity and its associated costs is critical to addressing the maternal health crisis in the United States. To develop effective maternal programming and policies and assess their impacts on outcomes, researchers and program staff will need strong data on the prevalence and incidence of maternal morbidity and associated inequities, including information on whether any changes observed over time are due to reporting changes or true increases or decreases in prevalence or incidence. Our study demonstrates ongoing gaps in measurement of maternal morbidity, however, and lack of data to identify associated disparities. Heath care and public health agencies, measure stewards, and health systems can do more to define measures beyond those for severe maternal morbidity and incorporate an equity lens to better understand the differential impact on various subgroups of the population. These subgroups for measurement can include individual demographics and community characteristics to provide further insight into social and structural conditions driving maternal outcomes. These analyses will support a systems-wide response to reducing maternal morbidity and addressing the underlying societal and structural causes.

## Supporting information

**S1 Appendix. Description of literature review and search terms.**
(DOCX)

**S2 Appendix. Prevalence of maternal morbidity conditions and outcomes.**
(DOCX)

**S3 Appendix. Studies and data sources used to inform the cost estimates used in the model.**
(DOCX)

**S4 Appendix. Evidence of the association between maternal morbidity conditions and maternal and child health outcomes.**
(DOCX)

**S5 Appendix. Model inputs: Parameters and costs used to estimate the economic impact of maternal morbidity conditions among 2019 births.**
(DOCX)

**S6 Appendix. Modeling method.**
(DOCX)

**S7 Appendix. Sensitivity analyses.**
(DOCX)

**S8 Appendix. Full results of the model.**
(DOCX)

**S9 Appendix. Glossary.**
(DOCX)

## Acknowledgments

The authors would like to thank Jodie Katon and Kay Johnson for their expert guidance in refining our conceptual model. We would also like to thank Caroline Margiotta and Jessica Gao for their support conducting the literature review, and Erin Lipman for her support in developing the Excel model. Finally, we thank all the researchers and practitioners for conducting and publishing the studies used in our review.

## Author Contributions

**Conceptualization:** Sasigant So O'Neil, Divya Vohra, Laurie Zephyrin, Kara Zivin.

**Funding acquisition:** Laurie Zephyrin.

**Investigation:** Isabel Platt, Eric Dehus.

**Methodology:** Kara Zivin.

**Project administration:** Sasigant So O'Neil, Eric Dehus.

**Software:** Emma Pendl-Robinson.

**Supervision:** Sasigant So O'Neil, Divya Vohra.

**Validation:** Isabel Platt, Emma Pendl-Robinson.

**Visualization:** Isabel Platt, Emma Pendl-Robinson.

**Writing – original draft:** Sasigant So O'Neil, Isabel Platt, Divya Vohra, Emma Pendl-Robinson.

**Writing – review & editing:** Isabel Platt, Laurie Zephyrin, Kara Zivin.

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
