## [Decision Letter · Decision Letter 0]

18 May 2022

PONE-D-22-11649Societal cost of nine selected maternal morbidities in the United StatesPLOS ONE

Dear Dr. ONeil,

Thank you for submitting your manuscript to PLOS ONE. After careful consideration, we feel that it has merit but does not fully meet PLOS ONE’s publication criteria as it currently stands. Therefore, we invite you to submit a revised version of the manuscript that addresses the points raised during the review process.

 The reviewers concurred on the importance of the project and generally approved the methods.  Their comments, particularly reviewer 2's comments, deserve thorough addressing.

We look forward to receiving your revised manuscript.

Kind regards,

Emily W. Harville

Academic Editor

PLOS ONE

Journal Requirements:

Reviewers' comments:

Reviewer's Responses to Questions

**Comments to the Author**

1. Is the manuscript technically sound, and do the data support the conclusions?

Reviewer #1: Yes

Reviewer #2: Yes

2. Has the statistical analysis been performed appropriately and rigorously? 

Reviewer #1: Yes

Reviewer #2: Yes

3. Have the authors made all data underlying the findings in their manuscript fully available?

Reviewer #1: Yes

Reviewer #2: Yes

4. Is the manuscript presented in an intelligible fashion and written in standard English?

Reviewer #1: Yes

Reviewer #2: Yes

5. Review Comments to the Author

Reviewer #1: Thank you for the opportunity to review this critically important and timely study. O’Neil et al. address an important policy and public health issue – societal costs of maternal morbidity. The paper is well written and applies appropriate methods to advance the topic. I have several suggestions and requests for points of clarification that I believe will help to strengthen the paper. These concerns and minor feedback are detailed below, by section:

Introduction:

- Line 13 - I suggest replacing 'developed' with high-income or industrialized countries.

- Line 25 - Is 2019 in the "maternal-pair from pregnancy to five years postpartum" referring to the end of postpartum period? It is confusing to read about the 2019 birth cohort and tracking the maternal-child pair 5 years postpartum (meaning to 2023?) - please clarify.

Methods:

- Line 93 - Could you please include a reference/citation to the Healthcare Cost and Utilization Project from which the Nationwide Inpatient Sample was drawn for those readers not familiar with it?

- Lines 126-128 - I am not convinced about the causal language used in key parameters 3 and 4 - does this mean you only selected those studies that estimated the impact (causal relationship)? Or how do you know the incidence of outcome (3) was caused by the maternal morbidity? Also, odds ratio estimates the association between morbidity-outcome, and not the impact of morbidity on outcome. I advise the authors to either clarify the criteria or adjust the language used to avoid inferences about the causality between morbidity-outcome.

Results:

- Table 1 - please include a brief explanation as to why some maternal morbidity outcomes have cost estimates for 1 year and other for up to 5 years?

Discussion:

- Lines 223-224 - I think the language like this has to be attenuated unless you only included those studies that looked at the causal effect and thus provide evidence that those outcomes truly resulted from / were caused by maternal morbidity?

- Limitations - I appreciate the detailed discussion (and acknowledgment) of study limitations!

S4 Appendix. Effects of Exposure to Maternal Morbidity Conditions – Again, I would suggest adjusting the language implying causality between maternal morbidity conditions and maternal and child health outcomes (e.g. “Evidence of the association between MMC and maternal and child health” or “MCH outcomes associated with MMC” )

I am impressed with the authors’ rigorous and transparent approach to documenting the conduct of this study and providing so much supporting information in the appendices. This is very helpful and greatly appreciated.

Reviewer #2: This is a meaningful contribution to the body of literature about maternal morbidity. Akin to the referenced publication by Luca et all assigning a cost to the impact of MMHC, this manuscripts attempts to estimate the cost of several maternal morbidities. This analysis addresses the question of how to assign economic metrics to maternal morbidity beyond the associated hospitalization and recognizes that these experiences have ramifications for maternal and early childhood health.

As an obstetric reader of this, I need a better explanation of why this list of 9 morbidities. AFE, Cardiac arrest, AKI, sepsis, VTE are all severe maternal morbidities. GDM, hemorrhage, HTN, MMHC are not. Do the categories of outcomes and predisposing condition perhaps deserve separate analyses? There should at minimum be an acknowledgement of the mixed list. It is also notable that the list includes AFE (estimated 1/40,000 births) and hypertension (1/5 births).

In the small proportion of outcomes that increase maternal costs in this analysis I don't see any reference to level of care -- use of intensive care rather than typical maternity postpartum stay, yet this would be an outcome of AFE, cardiac arrest, possibly hemorrhage, possibly sepsis.

There is a vulnerability in this model -- for example, hypertension is linked to cardiac arrest and renal disease, but the authors do not "amplify" the cost of hypertension by linking the outcomes attributed to these. An explanation for these omissions would be reasonable.

The opportunity to build on this to create more complex models that would include amplifications should be mentioned. This manuscript offers an excellent precedent and warrants publication with its analysis as it stands.

While the authors mention in their second to last paragraph that it would have been "ideal" to examine costs by race and ethnicity, they do not disclose why they didn't. It is true, as they mention, that SMM is more common in Black birthing populations, as are their other morbidities of interest (hemorrhage, GDM, hypertension). The compounded impact of lost productivity, income, mental health toll etc on a population already structurally marginalized may tell a very different story. Especially as there is data that MMHC are identified in non-hispanic white people at a greater rate than in hispanic or NHB populations due to underscreening of the latter -- that is, this analysis is vulnerable to structural racism due to the morbidities included: there are inherently more white patients in the MMHC group with massive cost associated with its sequelae, and more Black patients in the hypertension group with is nearly 3x lower cost. Perhaps in the interest of building the literature around cost of maternal morbidities this discrepancy can be acknolwedged not unpacked, but the manuscript deserves a better explanation of why race wasn't investigated.

6. PLOS authors have the option to publish the peer review history of their article (what does this mean?). If published, this will include your full peer review and any attached files.

Reviewer #1: No

Reviewer #2: No

---

## [Author Response · Author response to Decision Letter 0]

17 Jun 2022

Please see the "Response to Reviewers" letter included in this submission for a table outlining how we addressed each reviewer's suggestions.

---

## [Decision Letter · Decision Letter 1]

14 Sep 2022

PONE-D-22-11649R1Societal cost of nine selected maternal morbidities in the United StatesPLOS ONE

Dear Dr. ONeil,

Thank you for submitting your manuscript to PLOS ONE. After careful consideration, we feel that it has merit but does not fully meet PLOS ONE’s publication criteria as it currently stands. Therefore, we invite you to submit a revised version of the manuscript that addresses the points raised during the review process.

We thank you for your patience with the delays in reviews.  The reviewers agree the paper is substantially approved, but suggest a few minor clarifications. Please respond to these.

We look forward to receiving your revised manuscript.

Kind regards,

Emily W. Harville

Academic Editor

PLOS ONE

Journal Requirements:

Reviewers' comments:

Reviewer's Responses to Questions

**Comments to the Author**

1. If the authors have adequately addressed your comments raised in a previous round of review and you feel that this manuscript is now acceptable for publication, you may indicate that here to bypass the “Comments to the Author” section, enter your conflict of interest statement in the “Confidential to Editor” section, and submit your "Accept" recommendation.

Reviewer #1: All comments have been addressed

Reviewer #3: All comments have been addressed

2. Is the manuscript technically sound, and do the data support the conclusions?

Reviewer #1: Yes

Reviewer #3: Yes

3. Has the statistical analysis been performed appropriately and rigorously? 

Reviewer #1: Yes

Reviewer #3: Yes

4. Have the authors made all data underlying the findings in their manuscript fully available?

Reviewer #1: Yes

Reviewer #3: Yes

5. Is the manuscript presented in an intelligible fashion and written in standard English?

Reviewer #1: Yes

Reviewer #3: Yes

6. Review Comments to the Author

Reviewer #1: (No Response)

Reviewer #3: The authors have responded appropriately to the reviewers comments. A few points of clarification remain, and it would improve the paper if a little more clarity were added.

First, the study estimates costs over 5 years. Clearly the annual costs fall over this period, and it looks like most of the high costs consequences occur in this period. It would help if there were a clearer justification of the choice of 5 years, and a comment on the likely scale of underestimation of costs that result from this.

Second, i was a little unclear about the way in which health care inflation was used. There are two main reasons why costs in healthcare rise - general increases in pay and prices (which are not really relevant, since for decision making real rather than nominal costs are relevant), and increases in costs that are specific to the health sector, some of which are Baumol effects. The latter are relevant but are probably a small part of the increases.

Third, the co-morbidity point has been expanded, but i think we could still get a little more useful comment around this, particularly around the way in which costs for a single disease can be much higher in the context of co-morbidity.

Fourth, it is commented that the outcomes in question have risen substantially - is it clear if this is a real increase or a change in reporting?

Finally, and less seriously, it is a little odd to describe maternal deaths as one of the more serious outcomes. Some would argue it is the most serious!

7. PLOS authors have the option to publish the peer review history of their article (what does this mean?). If published, this will include your full peer review and any attached files.

Reviewer #1: No

Reviewer #3: No

---

## [Author Response · Author response to Decision Letter 1]

20 Sep 2022

Please see Response to Reviewers letter in the submission files.

---

## [Editor Report · Decision Letter 2]

21 Sep 2022

Societal cost of nine selected maternal morbidities in the United States

PONE-D-22-11649R2

Dear Dr. ONeil,

We’re pleased to inform you that your manuscript has been judged scientifically suitable for publication and will be formally accepted for publication once it meets all outstanding technical requirements.

Kind regards,

Emily W. Harville

Academic Editor

PLOS ONE
---

## [Editor Report · Acceptance letter]

28 Sep 2022

PONE-D-22-11649R2 

Societal cost of nine selected maternal morbidities in the United States 

Dear Dr. O’Neil:

I'm pleased to inform you that your manuscript has been deemed suitable for publication in PLOS ONE. Congratulations! Your manuscript is now with our production department. 

Kind regards, 

on behalf of

Dr. Emily W. Harville 

Academic Editor

PLOS ONE